# Swine intestinal segment perfusion model for the evaluation of nutrients bioaccessibility

**Matteo Dell'Anno** [ID]**, Fabio Acocella, Pietro Riccaboni, Camilla Recordati, Elisabetta Bongiorno, Luciana Rossi** [ID]*

Department of Veterinary Medicine and Animal Sciences - DIVAS, Università degli Studi di Milano, Lodi, Italy

* luciana.rossi@unimi.it

**Data Availability Statement:** All relevant data are available within the paper.

**Funding:** This research was financially supported by Lombardy Region (funding number: POR FESR

## Abstract

Nutrition science requires more science-based evidences for the development of effective functional diets. To reduce animals for experimental purposes innovative reliable and informative models, simulating the complex intestinal physiology, are needed. The aim of this study was to develop a swine duodenum segment perfusion model for the evaluation of nutrient bioaccessibility and functionality across time. At the slaughterhouse, one sow intestine was harvested following Maastricht criteria for organ donation after circulatory death (DCD) for transplantation purposes. Duodenum tract was isolated and perfused in sub-normothermic conditions with heterologous blood after cold ischemia induction. Duodenum segment perfusion model was maintained under controlled pressure conditions through extracorporeal circulation for 3 hours. Blood samples from extracorporeal circulation and luminal content samples were collected at regular intervals for the evaluation of glucose concentration by glucometer, minerals ($Na^+$, $Ca^{2+}$, $Mg^{2+}$, $K^+$) by ICP-OES, lactate-dehydrogenase and nitrite oxide by spectrophotometric methods. Dacroscopic observation showed peristaltic activity caused by intrinsic nerves. Glycemia decreased over time (from 44.00 ±1.20 mg/dL to 27.50±0.41; $p < 0.01$), suggesting glucose utilization by the tissue confirming the organ viability in line with histological examinations. At the end of the experimental period, intestinal mineral concentrations were lower than their level in blood plasma suggesting their bioaccessibility ($p < 0.001$). A progressive increase of LDH concentration over time was observed in the luminal content probably related to a loss of viability (from 0.32 ±0.02 to 1.36±0.02 OD; $p < 0.05$) confirmed by histological findings that revealed a de-epithelization of the distal portion of duodenum. Isolated swine duodenum perfusion model satisfied the criteria for studying bioaccessibility of nutrients, offering a variety of experimental possibilities in line with 3Rs principle.

## 1 Introduction

Nowadays, nutrition plays a pivotal role for both human and animal health, related not only to simply to satisfy nutritional requirements, it also plays a key role in the health and welfare also through its functionality [1–4]. The literature offers an heterogenous panorama of nutritional

2014–2020_BANDO Call HUB Ricerca e
Innovazione_D.G.R. NR 727 del 5 November
2018), within the project "MIND FoodS Hub
(Milano Innovation District Food System Hub):
Innovative concept for the eco-intensification of
agricultural production and for the promotion of
dietary patterns for human health and longevity
through the creation in MIND of a digital Food
System Hub", awarded to LR. This study was also
financially supported by the APC Central Fund of
the University of Milan. The funders had no role in
study design, data collection and analysis, decision
to publish, or preparation of the manuscript.

**Competing interests:** The authors declare that the
research was conducted in the absence of any
commercial or financial relationships that could be
construed as a potential conflict of interest.

studies, and several studies are contradictory on the argument. The prediction of the nutritional quality of food and feed products requires more knowledge related to the individual digestibility of dietary compounds [5].

Currently, the request to reduce animals for experimental purposes is continuously growing. European policies on animal experimentation are intensely aiming to increase the protection of experimental animals, thus various alternative methods have being developed to achieve this purpose [6]. Even the most advanced *in vitro* and *in silico* systems cannot fully simulate complex phenomena such as inflammation, digestion, pathologies, and metabolism. However, the introduction of innovative non-animal models is fundamental as complementary to animal experimentation [7]. Thus, reliable science-based models are required in order to reduce and replace animals for experimental purposes. The gastrointestinal physiology is a complex field that involves different tissues and systems (epithelium, muscles, nerves, connective, hormones and glandule), and the use of animals for studying nutrient digestion is still considered essential.

Though remarkable results achieved with *in vitro* models, organ architecture, epithelium integrity and nutrient absorption are not, at present, efficiently simulated. *In vitro* studies range in their level of complexity from simple monocultures to complex 3-D structures containing several cell types that are organized into a structure that retains (*ex vivo* tissue explants) or mimics (tissue engineered *in vitro* models) different *in vivo* elements [8]. The *ex vivo* digestion models are supposed to mimic the *in vivo* processes better than the *in vitro* technology, thanks to a manufacturing process of digestive fluids, and the presence of the complete array of proteases and substances that *in vivo* concur to the digestive process [9]. Several *ex vivo* studies have been conducted on pigs due to the similar characteristics of swine species in terms of similar morphology, structure, composition and enzymatic activity to humans and the low cost of this technology.

Organ explants efficiently simulate the entire animal complexity, offering the possibility to perform *ex vivo* experiments under standardized conditions and harvest several tissues from a single animal in line with the 3R principle (replace, reduce, refine). In addition, the possibility to perform more tests involving the same donor, improves the robustness of the *ex vivo* models [10]. It has been demonstrated that the mucosal epithelium is extremely susceptible to ischemia [11] and although *ex vivo* models can better reproduce the complexity of the *in vivo* situation when compared to *in vitro* models, deterioration of the explanted tissue and lack of hemodynamics can cause differences between *ex vivo* and *in vivo* data [12]. The perfusion of *ex vivo* tissue allow to the conservation of organ viability and structure for a longer period compared to classic *ex vivo* models [13]. Recent studies, showed encouraging results of swine intestinal perfused model suggesting it as a cost-effective, practical and reliable strategy for the study of intestinal physiology, pharmacology and transplantation [14]. In this scenario, the aim of this study was to develop and assess an innovative swine intestinal segment perfusion model for the evaluation of nutrient bioaccessibility and organ functionality across time for further applications to study the effect of feed ingredients and additives on nutrients bioavailability.

## 2 Materials and methods

### 2.1 Organ harvesting and extracorporeal circulation

One gastrointestinal tract was harvested at the slaughterhouse from a 100kg Large White sow, following the Maastricht criteria for organ donation after circulatory death (DCD) for transplantation purposes [15]. Heterologous blood was collected at the slaughterhouse and 25 IU of heparin and 1 g of Cefazolin were added to avoid coagulation and bacterial

contaminations. The blood was stored at 4 ˚C until organ perfusion. During the period of the "warm ischemia", that lasted from the animal exsanguination until the duodenum isolation, the mesenteric artery was cannulated [16]. The "cold ischemia" started with arterial infusion of 2 L of Belzer UW Cold Storage solution (S.A.L.F. Spa, Bergamo, Italy) [17] within 9 minutes for organ structure preservation. The descendant part of duodenum was isolated, and an intraluminal access was set-up by fixing a luer-lock connector on the pylorus side. The intestinal segment was intraluminal perfused with 360 mL of sterile Krebs-Ringer buffer (NaCl 115 mmol/L, $K_2HPO_4$ 2.4 mmol/L, $KH_2PO_4$ 0.4 mmol/L, $CaCl_2$ 1.2 mmol/L, $MgCl_2$ 1.2 mmol/L, $NaHCO_3^-$ 25 mmol/L, glucose 10 mmol/L) according to Biolley et al. [18] for favoring the epithelial structure conservation. The organ was transported to the laboratory submerged in a Ringer's lactate solution at 4˚C. Duodenum tract was maintained in controlled pressure conditions (flux pump 1.48 L/min, $O_2$ 2 L/min, artery pressure 76/55 mmHg) through extracorporeal circulation with heterologous blood at sub-normothermic perfusion condition for 3 hours starting by the addition of warm blood at 37 ˚C. The circuit was composed by a peristaltic pump (as a beating heart), venous reservoir and blood oxygenator (as lungs) [19]. Organ temperature was monitored using a thermometer for the entire experimental period.

## 2.2 Sample collection and analyses

During the first hour of extracorporeal circulation blood samples were collected from the line at 7 minutes interval (T0, T1, T2, T3, T4, T5, T6, T7, T8). Throughout the second hour, hematic samples were collected each 15 minutes (T9, T10, T11, T12) and at the third hour the collection was performed at 30 minutes intervals (T13, T14). Plasma was obtained by centrifugation at 3000 rpm for 15 min at 4 ˚C. In addition, the intestinal content was collected from T0 to T14 every 30 minutes from the luer-lock connector. All biological samples were evaluated for mineral and metabolites concentrations.

Glucose concentration was evaluated immediately after blood and intestinal lumen solution sampling from extracorporeal circulation line and the luer-lock connector using a glucometer (U-Right 4278, Biochemical Systems International S.p.A, Arezzo, Italy). The content of main macro-elements ($Na^+$, $K^+$, $Mg^{2+}$, $Ca^{2+}$) was assessed using Inductively Coupled Plasma Optical Emission Spectroscopy (ICP-OES, Optima 3300 XL, Perkin Elmer Inc., USA). First, calibration curves for each element considered were obtained using certified reference materials. Blood and intestinal content samples were then diluted 1:100 (v/v) with sterile MilliQ water, filtered with 0.45 um syringe filters and injected. The pH of intestinal lumen content was evaluated with pHmeter. Lactate dehydrogenase (LDH) and nitrite ion ($NO_2^-$) concentrations were evaluated through colorimetric kits (CytoTox 96® Non-Radioactive Cytotoxicity Assay and Griess Reagent System, Promega Italia S.r.l, Milan, Italy) following the manufacturer instructions. In particular, absorbances were read at 490 nm for LDH and 540 nm for $NO_2^-$ using a spectrophotometer (Model 680 Microplate Reader; Bio-Rad Laboratories, Inc., Hercules, CA, USA).

## 2.3 Histology, immunohistochemistry, and digital image analysis

After 3 hours of continuous perfusion, the intestinal segment was subdivided in three portions (proximal A, medial B, and distal C; Fig 1) and tissue samples were fixed in 10% neutral buffered formalin, embedded in paraffin, sectioned at 4 µm, and stained with hematoxylin and eosin (H&E).

To assess the extent of epithelial cells proliferation in the intestinal mucosa, immunohistochemistry with anti-ki67 (SP6, RM-9106-S, Labvision) primary antibody was performed.

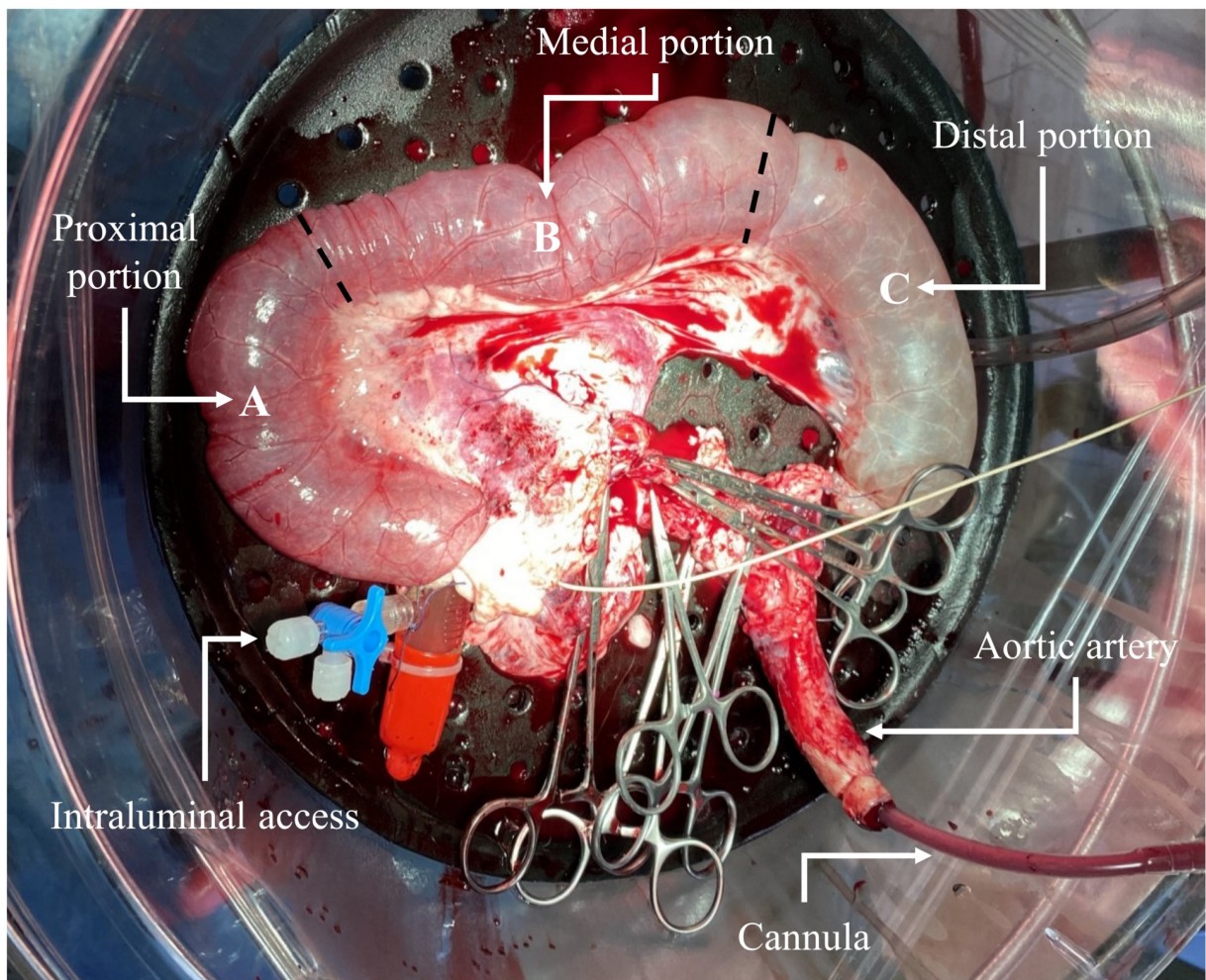

**Fig 1. Representative image of the intestinal segment perfusion model, ideally subdivided in proximal (A), medial (B) and distal (C) portions for tissue sampling.** The outlined lines indicate the incision points.

Four μm sections were deparaffinized and underwent heat-induced epitope retrieval at pH 9, for 40 min at 95°C (Dewax and HIER Buffer H, TA-999-DHBH, Thermo Scientific, UK). Endogenous peroxidase activity was blocked by incubating sections in 3% $H_2O_2$ for 10 min. Slides were rinsed and treated with PBS containing 10% normal serum for 30 min to reduce nonspecific background staining and then incubated for 1 hour at room temperature with the primary antibodies. Sections were incubated for 30 min with appropriate biotinylated secondary antibodies (Vector Laboratories, Burlingame, CA, USA), and then labelled by the avidin-biotin-peroxidase (ABC) procedure with a commercial immunoperoxidase kit (VECTAS-TAIN Elite ABC HRP Kit Standard, PK-6100, Vector Laboratories, Burlingame, CA, USA). The immunoreaction was visualized with 3,3'-diaminobenzidine substrates (Peroxidase DAB Substrate Kit, VC-SK-4100-KI01, Vector Laboratories, Burlingame, CA, USA) for 5 min and sections were counterstained with Mayer's hematoxylin. Digital slides were obtained from H&E, and immunostained sections by using the NanoZoomer-S60 Digital slide scanner (Hamamatsu, C13210-01), and images were captured by using theNDP.view2 Viewing software (Hamamatsu, U12388-01).

To assess the proliferation of the intestinal mucosa, the % of ki67-positive epithelial cells (number of ki67 positive nuclei/number of total nuclei x 100) was evaluated using the ImageJ analysis program [20].

## 2.4 Statistical analysis

The results of blood and intestinal content glucose, minerals and metabolites were analyzed using a one-way ANOVA using JMP® Pro 15 (SAS Inst. Inc., Cary, NC, USA). Multiple comparisons among timepoints were evaluated by performing Tukey's Honestly Significant Difference test (Tukey's HSD). The results were presented as least squares means ± standard error (SE). The means were considered different when $p \leq 0.05$.

# 3 Results

During the extracorporeal circulation, pressure and oxygenation conditions were maintained stable showing no alterations for three hours. Organ temperature was progressively increasing over time, maintaining the sub-normothermic perfusion throughout the experimental period (T0 = 15.2 ˚C, T1 = 17.5 ˚C, T2 = 18.8 ˚C, T3 = 19.2 ˚C, T4 = 20.5 ˚C, T5 = 21.0 ˚C, T6 = 22.0 ˚C, T7 = 23.0 ˚C, T8 = 23.6 ˚C, T9 = 24.5 ˚C, T10 = 25.0 ˚C, T11 = 25.0 ˚C, T12 = 25.6 ˚C, T13 = 25.6 ˚C, T14 = 25.8 ˚C). In general, the intestinal segment showed redness due to blood vessels reperfusion, except in the distal portion C that appeared pale and bloodless, and the peristaltic activity was observed after pinching stimulation. Reaching 25.3 ˚C the peristaltic contraction was noted without any external stimulation.

## 3.1 Glucose, mineral concentrations, organ temperature and intestinal pH

The glucose level measured in Krebs-Ringer solution was 123.67±2.31 mg/dL. The glycemia revealed a constant decreasing trend from an initial concentration of 44.00±1.20 to 27.50±0.41 mg/dL after 3 hours of extracorporeal circulation (Fig 2).

Considering the sampling intervals, during the first hour (T0-T8) statistically significant different concentrations were registered when comparing T0 to T5, T6, T7 and T8 (44.00 ±1.20, 36.50±1.20, 41.50±1.20, 38.25±1.20, 38.50±1.20 mg/dL respectively; ($p < 0.001$). In the course of the second (T8-T12) and third hour (T12-T14) glucose concentration gradually decreased until the end of experiment ($p < 0.01$). The glucose concentrations of intestinal lumen did not significantly differ from the beginning to the end of extracorporeal circulation (T0: 94.00±6.43 mg/dL; T4: 99.50±6.43 mg/dL; T8: 97.50±6.43; T12: 90.50±4.55 mg/dL; T14: 84.00±5.25).

Considering the mineral concentrations, $Ca^{2+}$, $Mg^{2+}$, $K^+$ and $Na^+$ showed significant differences over the experimental time (Fig 3). In particular, $Na^+$ blood levels raised from T0 to T4 and decreased slightly at T14, while its concentration in intestinal content was constantly reducing over time ($p < 0.01$). $Ca^{2+}$, $Mg^{2+}$, $K^+$ blood concentrations increased significantly over time, and co-currently their concentrations in the intestinal lumen significantly dropped from T0 to T14 ($p < 0.01$).

Intestinal pH showed a constant value of 7 registered for T0, T4, T8, T12 and T14.

## 3.2 Lactate dehydrogenase and nitrite concentrations

LDH blood concentrations revealed a persistent trend over time even if statistically significant differences were observed only in T3 and T4 compared to T14 (T3: 0.77±0.05 OD, T4: 0.85 ±0.05 OD and T14: 1.08±0.05 OD; $p < 0.05$; Table 1). Intestinal lumen content registered raising concentrations of LDH from T4 to T14 ($p < 0.0001$).

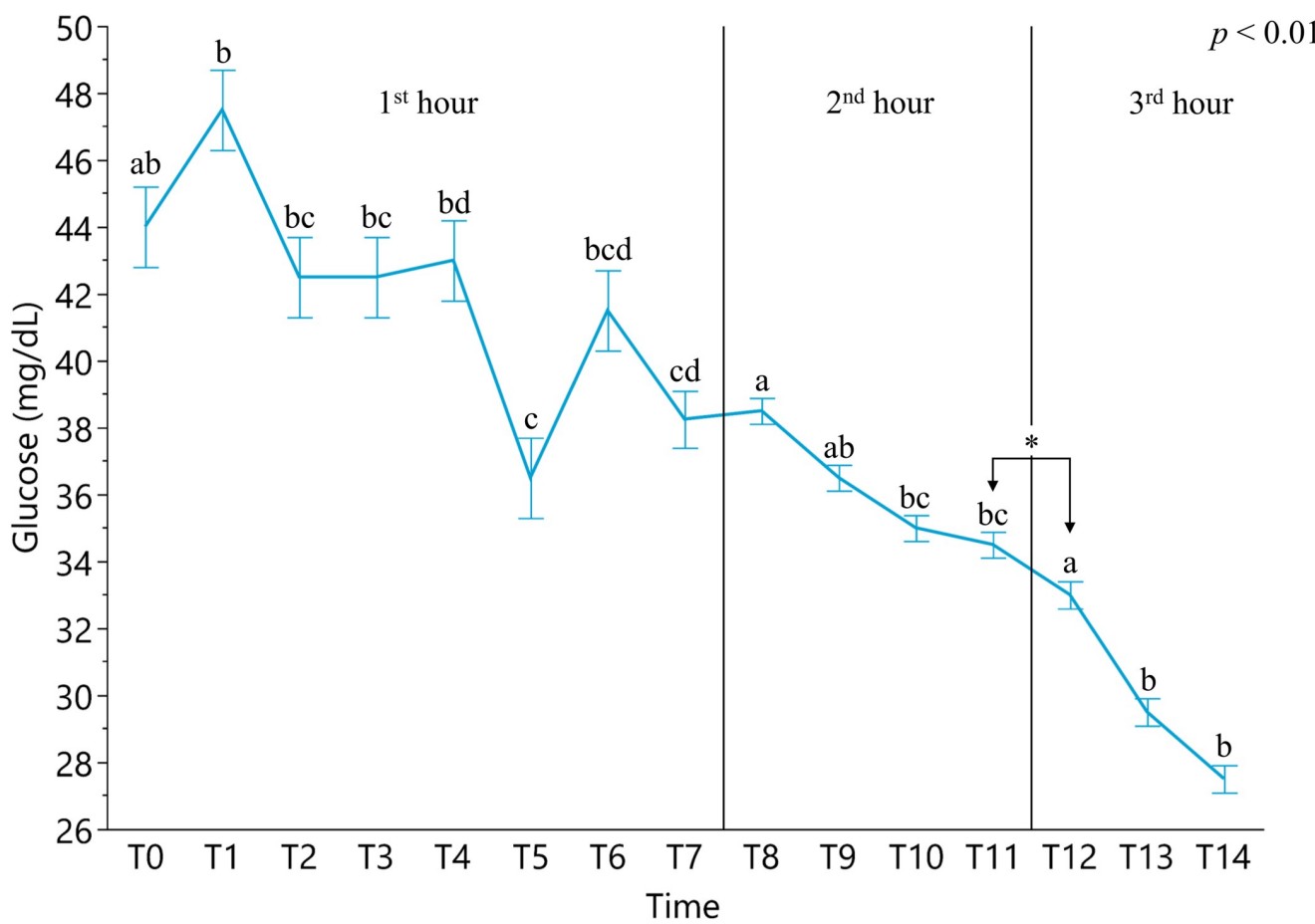

**Fig 2. Glycemic levels measured for 3 hours from the extracorporeal circulation of perfused intestinal segment.** Data are expressed as least squared means and standard errors. Different lowercase letters indicate statistically significant differences ($p \leq 0.01$). First hour: hematic glucose concentrations of the first hour measured at 7 minutes intervals; Second hour: hematic glucose concentrations of the second hour measured at 30 minutes intervals; Third hour: hematic glucose concentrations of the third hour measured at 30 minutes intervals. Asterisk indicates statistically significant different values.

Nitrite ion concentrations were below the detection level of the assay for the entire experimental period ($< 1.56\ \mu M$).

### 3.3 Histological evaluation of the intestinal segment

In all examined portions (A, B, C), the intestinal architecture was preserved. A diffuse moderate infiltration of lymphocytes, plasma cells and lesser numbers of neutrophils was present in the intestinal mucosa, consistent with a diffuse moderate subacute enteritis. Moderate hyperemia was present in portions A and B (Fig 4). The intestinal epithelium of A and B portions were similar in terms of a slight de-epithelialization in the apical part of villi (Fig 4A and 4B). On the contrary, the intestinal portion C showed absence of erythrocytes within blood vessels (impaired perfusion) and a diffuse necrosis of villi was visible (Fig 4C). Immunohistochemistry analysis revealed 47.58% of proliferating epithelial cells in portion A, 35.01% in portion B and 33.72% in portion C (Fig 5).

### 4 Discussion

This study aimed to investigate the possibility to develop an alternative perfused intestinal segment model to study the nutrient bioaccessibility and bioavailability targeting to provide more

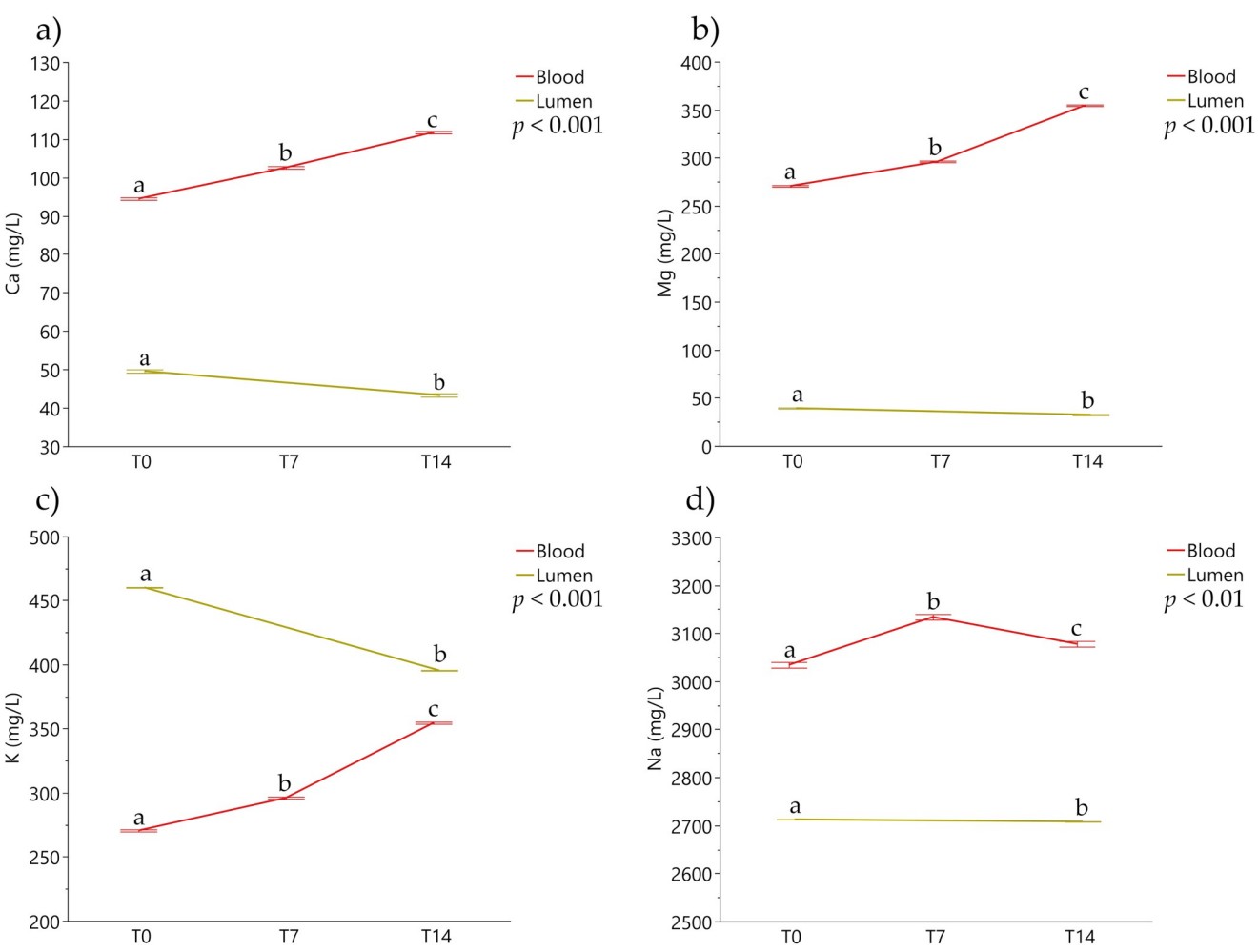

**Fig 3. Mineral concentrations measured over time in blood plasma and intestinal lumen content.** Data are expressed as least squared means and standard errors. Different lowercase letters indicate statistically significant differences ($p \leq 0.05$). a) Calcium ($Ca^{2+}$) concentrations measured at T0-T7-T14 in blood and T0-T14 in intestinal lumen content; b) Magnesium ($Mg^{2+}$) concentrations measured at T0-T7-T14 in blood and T0-T14 in intestinal lumen content; c) Potassium ($K^+$) concentrations measured at T0-T7-T14 in blood and T0-T14 in intestinal lumen content; d) Sodium ($Na^+$) concentrations measured at T0-T7-T14 in blood and T0-T14 in intestinal lumen content.

complete data compared to *in vitro* models. It is important to consider that this model is more advanced if compared to a classic *ex vivo* model which do not provide extracorporeal circulation. This model was obtained following the surgery techniques used for *in vivo* organ transplantations in order to preserve the organ viability and architecture during the entire experimental period [13].

Glucose bioaccessibility was evaluated due to its relevance *in vivo* as energetic marker derived from starch and sugar digestion. The glucose level is considered a marker of cell proliferation, viability [21, 22] and functionality for the metabolic activity [23]. The initial glycemic level (44.00±1.20 mg/dL) could be considered low if compared to the normal range of swine species (from 80 to 120 mg/dL) [24, 25]. The observed initial hypoglycaemia could be due to the animal fasting before slaughtering procedures. The glycemic trend showed a decreasing curve suggesting the utilization of glucose by erythrocytes and intestinal cells and the impossibility to store sugars in the intestinal tissue after blood perfusion. We observed a peak of glucose absorption at 7 minutes of extracorporeal circulation, and after the first hour a drop of

**Table 1. Lactate dehydrogenase concentrations measured over time in blood and intestinal lumen content.**

| Timepoint | Blood (OD) | Lumen (OD) |
|---|---|---|
| T0 | 0.93±0.05[ab] | 0.32±0.02[a] |
| T1 | 0.99±0.05[ab] | |
| T2 | 0.81±0.05[ab] | |
| T3 | 0.77±0.05[a] | |
| T4 | 0.84±0.05[a] | 0.34±0.02[a] |
| T5 | 0.87±0.05[ab] | |
| T6 | 0.82±0.05[ab] | |
| T7 | 0.89±0.05[ab] | |
| T8 | 0.89±0.05[ab] | 0.50±0.02[b] |
| T9 | 0.86±0.05[ab] | |
| T10 | 0.92±0.05[ab] | |
| T11 | 0.83±0.05[ab] | |
| T12 | 0.92±0.05[ab] | 0.80±0.02[c] |
| T13 | 1.01±0.05[ab] | |
| T14 | 1.08±0.05[b] | 1.36±0.02[d] |
| *p-value* | 0.0256 | < 0.0001 |

Data are expressed as least squared means and standard errors.

Different lowercase letters indicate statistically significant differences ($p \leq 0.05$).

OD: optical densities measured at 490 nm of wavelength.

18% was registered. Even without registering significant differences, endoluminal content of glucose showed decreasing concentrations probably due to the low amount of glucose consumed by erythrocytes for their metabolism [26]. However, the observed results suggested that glucose was transported through its traditional mechanism. Glucose uptake is mediated via the Sodium-Glucose Cotransporter 1 (SGLT1) localized on the enterocytes membrane and its basolateral transport to the blood circulation is mediated primarily by the Glucose Transporter 2 (GLUT2) [27]. Glucose absorption is a complex mechanism also mediated by pancreas and renin-angiotensin-aldosterone [28]. In this study, the glucose absorption could not be mediated by hormonal asset [29] since it can be fully assessed only *in vivo*. The glucose concentration used was based on the defined level for the Krebs-Ringer solution as intraluminal nutrient medium. The low intraluminal glucose concentration [30] confirmed the organ metabolism without registering osmotic damage [31], even if it was not possible to evaluate a kinetic curve

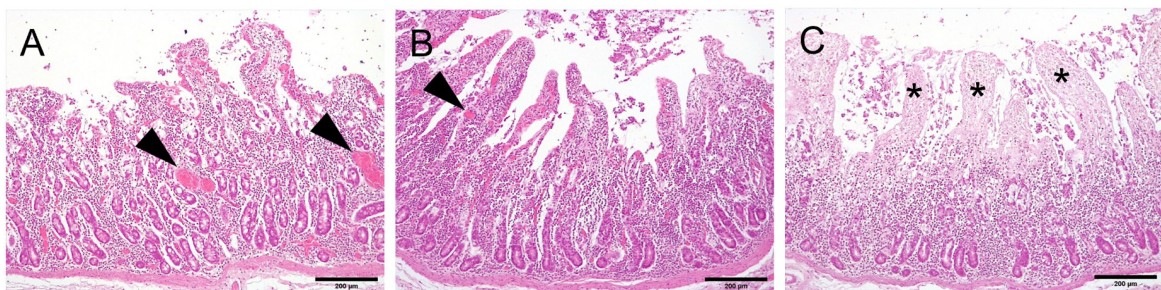

**Fig 4. Histology of the perfused intestinal tract at the end of the 3 hours of extracorporeal circulation (H&E stain, 100x, scale bar = 200 μm).** Portions A and B were overall well preserved and perfused, as demonstrated by blood vessels engorged with erythrocytes (arrowheads). In portion C, blood vessels are not evident (impaired perfusion) and necrosis of villi (*) was present. Hematoxylin eosin of intestinal A, B and C portions.

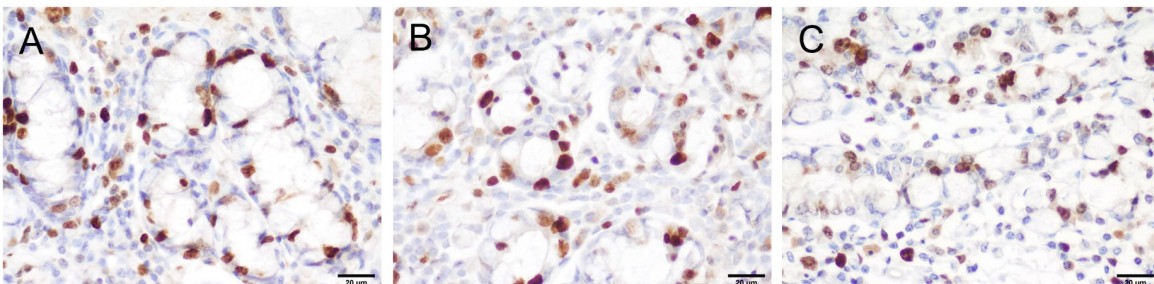

**Fig 5. Proliferation of intestinal crypt epithelial cells.** Immunoperoxidase staining for ki67, 400x, scale bar = 20 µm in portions A, B, and C.

of its uptake [14]. Observed results suggest that glucose could be considered an interesting marker for further application of this perfusion model for the evaluation of functional feed additives which are meant to improve gut health and nutrient utilization [32, 33].

Minerals are essential inorganic nutrients that have to be exclusively acquired from the diet [34]. Minerals concentration was evaluated as indirect indicator of intestinal function as they do not require digestion to be absorbed. $Na^+$ concentration in blood displayed a significant increase from T0 to T7 and decreased from T7 to T14. In the meantime, the intestinal lumen content of $Na^+$ dropped significantly from T0 to T14. The observed increase in osmolarity suggested that $Na^+$ uptake and utilization was maintained during the extracorporeal circulation. The absorption of $Na^+$ is associated with glucose uptake from Sodium-Glucose Transporters (SGLT1) [35]. In this model, the physiologic separation of vascular, interstitial, and intracellular sections was conserved, and the blood concentration of $Na^+$ required to maintain a correct balance of its level in the interstitial matrix.

$K^+$ is one of the most important minerals for the acid-base and osmotic pressure balance [36]. $K^+$ is the most abundant intracellular cation and its plasma level is lower than sodium abundance [37]. Similar to what observed for $Na^+$, $K^+$ levels increased in blood and decreased in lumen content from T0 to T14.

$Mg^{2+}$ is one of widely abundant minerals in the animal body, and it is involved in several pivotal processes such as energy production, muscular contraction and nervous impulses transferring [38]. The observed increased $Mg^{2+}$ plasma concentration suggests its absorption from intestinal lumen. In our study, $Mg^{2+}$ blood level showed a drop from T0 to T7 probably due to epithelial utilization of this mineral. Subsequently, the $Mg^{2+}$ concentration increased from T7 to T14 indicating its absorption from the lumen content. The intestinal lumen levels of $Mg^{2+}$ significantly dropped from T0 to T14 indicating a duodenal uptake through passive and facilitated diffusion processes [39]. $Ca^{2+}$ absorption is achieved through the active transport and passive diffusion [40]. Its absorption is related to the $Mg^{2+}$ presence, involving the ATP-dependent ionic pumps that transfer $Ca^{2+}$ in the extracellular space exchanging calcium with $Na^+$ [41]. $Mg^{2+}$ could be also indirectly influenced by $Na^+$ and $K^+$ concentrations since it is involved in the activity of the sodium-potassium pump [42]. Gastrointestinal system and kidneys closely regulate $Mg^{2+}$ absorption and elimination [43], and its intestinal active and passive uptake mechanisms seem to do not be under hormonal control [44]. For this reason, $Mg^{2+}$ concentration could be considered a translational parameter to *in vivo*, that indicates the nutrients' bioaccessibility in the following perfusion model.

In line with the registered trend of $Mg^{2+}$, $Ca^{2+}$ levels significantly raised in blood and dropped in the intestinal lumen content. $Ca^{2+}$, $Mg^{2+}$ and $K^+$ plasma concentrations showed a decrease in case of enteritis *in vivo* which can impair the absorption functionality of the gut

epithelium [45, 46]. In our duodenum perfusion segment model, luminal and plasma mineral concentrations demonstrated that the bioaccessibility of these nutrients was maintained over three hours of extracorporeal circulation.

The lactate dehydrogenase is a cytosolic enzyme that catalyzes the conversion of lactate to pyruvate. LDH measurements offer a non-invasive and objective indication of mucosal cellular integrity since the extracellular localization of LDH is associated with an epithelial tissue injury [47]. LDH blood concentrations did not show any difference over time even if higher numerical values were registered after 3 hours. During an organ injury LDH is released from dead cells and its concentration increases both in plasma and intestinal content [48]. In addition, LDH is strongly correlated with hemoglobin release in blood vessels due to hemolysis [49]. The registered levels of LDH in plasma suggests that peristaltic pump and pressure conditions used for extracorporeal circulation were suitable in terms of erythrocytes integrity [50]. The perfused intestinal segment model showed a progressive increase of luminal concentration of LDH suggesting a progressive damage of epithelial mucosa, consistent with the intestinal de-epithelialization and necrosis observed during the histological evaluation.

In this model, LDH concentrations in plasma and intestinal lumen content could be considered as reliable indirect marker to assess organ viability over time.

In line with Sundaram et al. [51], nitric oxide was not detectable, probably due to the limited ability of swine enterocytes to produce $NO_2^-$.

The preservation of tissue integrity before and during the experimentation is pivotal to ensure the accuracy and reproducibility of data [52]. Histological evaluation revealed that the overall intestinal architecture was preserved after 3 hours, with a visible blood perfusion and only a slight de-epithelialization of villi in portions A and B, while in portion C perfusion was absent and a diffuse loss of intestinal villi was observed. In this distal portion the nutrient absorption was likely impaired. Intestinal proliferation was highest in proximal portion A with a progressive decrease in B and C, supporting a reduced vitality in particular in the distal portion C. In the examined portions of duodenum, the observed enteritis was considered a spontaneous finding not unexpected in a farming pig. The progressive impairment of section C structure and tissue death could lead to a gradual necrosis that could involve the entire intestinal segment if the perfusion time had been extended. Preservation of epithelial structures for 3 hours was proposed as the optimal period to optimize organ viability in *ex vivo* studies [53]. The observed increase in LDH was likely due to the de-epithelization of the intestinal mucosa. Hyperemia observed in the portions A and B was probably due to high pressure conditions, even if the selected parameters were in accordance with previous studies. Future development of the following model will target to ensure a complete perfusion by shortening the segment and perfection the surgery technique in order to store the complete vessels' architecture that could have influenced the blood perfusion in distal section. In this model, pressure conditions require to be adapted in accordance with the size of the considered intestinal segment, and this aspect will require further adaptations in order to perfection this model. Literature studies demonstrated that UW solution provides better conservation of patches through prolonged ischemia compared to other solutions [54]. Further studies are needed to assess the glucose absorption trough perfusion of higher glucose concentrations through the intraluminal line. In addition, more data are required to investigate and optimize the luminal nutrition in order to maximize the organ viability and the conservation of intestinal structure for longer periods.

## 5 Conclusions

The developed swine perfused intestinal segment model showed characteristics of organ viability and functionality over three hours of extracorporeal circulation. The duodenum segment

preserved the principal nutrients bioaccessibility, cellular metabolism, and, at histological evaluation, showed a preserved intestinal epithelial lining in the perfused portions. This study offered encouraging results for the development of a novel swine intestinal segment perfusion model for the evaluation of nutrients bioaccessibility in line with the 3R principle. Future studies will be useful to improve the organ viability and structures conservation, considering the potential of this model also for application to translational medicine for intestinal transplantations.

## Acknowledgments

The authors acknowledge the support of the APC central fund of the University of Milan.

## Author Contributions

**Conceptualization:** Fabio Acocella, Luciana Rossi.

**Data curation:** Matteo Dell'Anno.

**Formal analysis:** Matteo Dell'Anno, Fabio Acocella, Pietro Riccaboni, Camilla Recordati, Elisabetta Bongiorno, Luciana Rossi.

**Funding acquisition:** Luciana Rossi.

**Investigation:** Matteo Dell'Anno, Fabio Acocella, Pietro Riccaboni, Camilla Recordati, Elisabetta Bongiorno, Luciana Rossi.

**Methodology:** Matteo Dell'Anno, Fabio Acocella, Pietro Riccaboni, Camilla Recordati, Elisabetta Bongiorno, Luciana Rossi.

**Supervision:** Fabio Acocella, Luciana Rossi.

**Visualization:** Matteo Dell'Anno, Pietro Riccaboni, Camilla Recordati.

**Writing – original draft:** Matteo Dell'Anno, Fabio Acocella.

**Writing – review & editing:** Matteo Dell'Anno, Fabio Acocella, Pietro Riccaboni, Camilla Recordati, Luciana Rossi.

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
