## [Decision Letter · Decision Letter 0]

13 Mar 2023

PONE-D-22-35014Swine intestinal segment perfusion model for the evaluation of nutrients bioaccessibilityPLOS ONE

Dear Dr. Rossi,

Thank you for submitting your manuscript to PLOS ONE. After careful consideration, we feel that it has merit but does not fully meet PLOS ONE’s publication criteria as it currently stands. Therefore, we invite you to submit a revised version of the manuscript that addresses the points raised during the review process.

ACADEMIC EDITOR: Minor revision. The reviewers have recommended the possibility of a publication, but suggest some revisions to your manuscript. I kindly invite you to respond to the reviewers' comments and revise your manuscript accordingly.

We look forward to receiving your revised manuscript.

Kind regards,

Ozlem Boybeyi-Turer

Academic Editor

PLOS ONE

Journal Requirements:

2. "PLOS requires an ORCID iD for the corresponding author in Editorial Manager on papers submitted after December 6th, 2016. Please ensure that you have an ORCID iD and that it is validated in Editorial Manager. To do this, go to ‘Update my Information’ (in the upper left-hand corner of the main menu), and click on the Fetch/Validate link next to the ORCID field. This will take you to the ORCID site and allow you to create a new iD or authenticate a pre-existing iD in Editorial Manager. Please see the following video for instructions on linking an ORCID iD to your Editorial Manager account: " ext-link-type="uri" xlink:type="simple">https://www.youtube.com/watch?v=_xcclfuvtxQ"

Reviewers' comments:

Reviewer's Responses to Questions

**Comments to the Author**

1. Is the manuscript technically sound, and do the data support the conclusions?

Reviewer #1: Yes

Reviewer #2: Yes

2. Has the statistical analysis been performed appropriately and rigorously? 

Reviewer #1: Yes

Reviewer #2: Yes

3. Have the authors made all data underlying the findings in their manuscript fully available?

Reviewer #1: Yes

Reviewer #2: Yes

4. Is the manuscript presented in an intelligible fashion and written in standard English?

Reviewer #1: Yes

Reviewer #2: Yes

5. Review Comments to the Author

Reviewer #1: This paper decribes an attempt to use a porcine intestinal segment to use as an ex vivo model for nutrient bioavailability. It is well though of and competently written study with some limitations that are however pointed out by the authors. These limitations derive from the fact that this is considered a preliminary study that should be followed by additional research. My main criticism that I want to be addressed by the authors is there is a lack of discussion on the potential effect on the reported results of the de-epithelization and necrosis observed in segment C. The authors should also hypothesize its effect across time since this is a time-dependent process. Also they should be more specific in the last paragraph before the conclusion section (L282-284) considering their future approach steps would they consider to maximize organ viability and intestinal structure. Finally I have made several corrections/suggestions to improve the final manuscript and pointed out some sentences that need rephrasing to make them clearer. These minor comments are included as cooments using the adobe acrobat tools in the attached file.

Reviewer #2: The European policies on animal experimentation are aiming to increase the protection of experimental animals, thus various alternative methods have to be developed to achieve this purpose. Thus, reliable science based models are required in order to reduce and replace animals for experimental purposes.

In this study, a perfusion model of the porcine duodenal segment was developed for the evaluation of the bioaccessibility and functionality of nutrients over time. The duodenum segment showed characteristics of vitality and functionality of the organ over three hours of extracorporeal circulation. It also preserved the bioaccessibility of the main nutrients, the cellular metabolism and, on histological evaluation, showed a preserved intestinal epithelial lining in the perfused portions.

This study offered encouraging results for the development of a novel porcine intestinal segment perfusion model for the assessment of nutrient bioaccessibility in line with the 3Rs principle.

The manuscript is well written and well organized. The information presented are new and interesting and the conclusions are supported by the data. The Introduction introduces the topic sufficiently. The methodology is sufficiently detailed to allow the experiments to be reproduced. The results of this work will be very useful for further studies and they were also appropriately discussed. Finally, the quality of the English is adequate.

In my opinion the manuscript is acceptable for publication.

6. PLOS authors have the option to publish the peer review history of their article (what does this mean?). If published, this will include your full peer review and any attached files.

Reviewer #1: No

Reviewer #2: No

---

## [Author Response · Author response to Decision Letter 0]

16 Mar 2023

PONE-D-22-35014

Swine intestinal segment perfusion model for the evaluation of nutrients bioaccessibility

PLOS ONE

Dear Dr. Rossi,

Thank you for submitting your manuscript to PLOS ONE. After careful consideration, we feel that it has merit but does not fully meet PLOS ONE’s publication criteria as it currently stands. Therefore, we invite you to submit a revised version of the manuscript that addresses the points raised during the review process.

ACADEMIC EDITOR: Minor revision. The reviewers have recommended the possibility of a publication, but suggest some revisions to your manuscript. I kindly invite you to respond to the reviewers' comments and revise your manuscript accordingly.

5. Review Comments to the Author

Reviewer #1: This paper decribes an attempt to use a porcine intestinal segment to use as an ex vivo model for nutrient bioavailability. It is well though of and competently written study with some limitations that are however pointed out by the authors. These limitations derive from the fact that this is considered a preliminary study that should be followed by additional research. My main criticism that I want to be addressed by the authors is there is a lack of discussion on the potential effect on the reported results of the de-epithelization and necrosis observed in segment C. The authors should also hypothesize its effect across time since this is a time-dependent process. Also they should be more specific in the last paragraph before the conclusion section (L282-284) considering their future approach steps would they consider to maximize organ viability and intestinal structure. Finally I have made several corrections/suggestions to improve the final manuscript and pointed out some sentences that need rephrasing to make them clearer. These minor comments are included as cooments using the adobe acrobat tools in the attached file.

Authors: Thank you so much for your constructive discussion. The progressive impairment of section C structure and tissue death could lead to a gradual necrosis that could involve the entire organ. The future aim for the enhancement of the following model will target to ensure a complete perfusion by shortening the segment and perfection the surgery technique in order to store vessels’ architecture that could have influenced the blood perfusion in distal section (C) (Lines 274-282). The paper has been revised according to your comments/corrections. Find our response point-by-point to your comments in the attached PDF file.

Reviewer #2: The European policies on animal experimentation are aiming to increase the protection of experimental animals, thus various alternative methods have to be developed to achieve this purpose. Thus, reliable science based models are required in order to reduce and replace animals for experimental purposes.

In this study, a perfusion model of the porcine duodenal segment was developed for the evaluation of the bioaccessibility and functionality of nutrients over time. The duodenum segment showed characteristics of vitality and functionality of the organ over three hours of extracorporeal circulation. It also preserved the bioaccessibility of the main nutrients, the cellular metabolism and, on histological evaluation, showed a preserved intestinal epithelial lining in the perfused portions.

This study offered encouraging results for the development of a novel porcine intestinal segment perfusion model for the assessment of nutrient bioaccessibility in line with the 3Rs principle.

The manuscript is well written and well organized. The information presented are new and interesting and the conclusions are supported by the data. The Introduction introduces the topic sufficiently. The methodology is sufficiently detailed to allow the experiments to be reproduced. The results of this work will be very useful for further studies and they were also appropriately discussed. Finally, the quality of the English is adequate.

In my opinion the manuscript is acceptable for publication.

Authors: We are grateful for the time that you dedicated to our manuscript, and we are glad that the submitted paper could be suitable for publication.

---

## [Editor Report · Decision Letter 1]

20 Mar 2023

Swine intestinal segment perfusion model for the evaluation of nutrients bioaccessibility

PONE-D-22-35014R1

Dear Dr. Rossi,

We’re pleased to inform you that your manuscript has been judged scientifically suitable for publication and will be formally accepted for publication once it meets all outstanding technical requirements.

Kind regards,

Ozlem Boybeyi-Turer

Academic Editor

PLOS ONE
---

## [Editor Report · Acceptance letter]

6 Apr 2023

PONE-D-22-35014R1 

Swine intestinal segment perfusion model for the evaluation of nutrients bioaccessibility 

Dear Dr. Rossi:

I'm pleased to inform you that your manuscript has been deemed suitable for publication in PLOS ONE. Congratulations! Your manuscript is now with our production department. 

Kind regards, 

on behalf of

Professor Ozlem Boybeyi-Turer 

Academic Editor

PLOS ONE